# Assessing the Sustainability of Inflation Targeting: Evidence from EU Countries with Non-EURO Currencies

**Adina Ionela Străchinaru [1] and Bogdan Andrei Dumitrescu [2],***

[1] Department of Money and Banking, Bucharest University of Economic Studies, 010961 Bucharest, Romania; adina.strachinaru@fin.ase.ro
[2] Department of Money and Banking and Center of Financial and Monetary Research CEFIMO, Bucharest University of Economic Studies, 010961 Bucharest, Romania
* Correspondence: bogdan.dumitrescu@fin.ase.ro

**Abstract:** This paper examines the impact of inflation targeting (IT) adoption on macroeconomic outcomes (unemployment, inflation, exchange rate, and its volatility) and banking concentration by comparing the non-EUR European countries that adopt an IT strategy to non-EUR European countries that do not adopt an IT strategy for the period of 2005–2015. The results suggest that IT has no impact on inflation, unemployment, and the exchange rate raises the systemic risk. Moreover, in non-IT countries, central banks are more concerned to minimizing the exchange rate volatility to better protect the debtors with foreign currency (EURO) loans.

**Keywords:** inflation targeting; exchange rate volatility; financial stability; central banks

## 1. Introduction

Inflation targeting (IT) has become a very popular strategy since 1989, when the Reserve Bank of New Zealand used this pioneering framework to win the fight against the major macroeconomic disequilibrium represented by inflation. Given its flexibility, over the last three decades, more and more countries have chosen this path to conduct their monetary policy. The debates on this topic have led to five criteria characterizing IT—(i) the public announcement of inflation target, (ii) central bank engagement to price stability as its main goal, (iii) forward-looking strategy in respect with inflation predictions, (iv) increased transparency of the operations carried out by the central banks, and (v) higher accountability of the central bank when it comes to achieving its objectives.

Leaving aside its apparent complexity, IT strategy is straightforward. The central banks estimate the evolution of inflation in the near future and compare it with the target they have established based on current macroeconomic conditions and ultimately decide how much monetary policy needs to be calibrated. Some central banks have utilized symmetrical intervals around a midpoint for inflation target, while other central banks have proposed only a reference rate or an upper limit to inflation.

Even though price stability is the main objective of most central banks worldwide, with or without an IT regime, they also pursue some secondary goals such as preserving financial stability while maximizing employment and economic growth [1]. More to the point, central banks oversee the banking system to make sure that commercial banks are financially sustainable and following, at the same time, thoughtful management practices. For this reason, investigating how an IT framework interacts with some additional goals from a given central bank's agenda is a matter of great importance, since implementing this strategy involves a trade-off between inflation and other macroeconomic variable like unemployment.

In this paper, we examine the impact of IT adoption on macroeconomic outcomes by comparing the non-EUR European countries which adopt an IT strategy to non-EUR European countries that did not adopt an IT strategy for the period of 2005–2015. Based on the Propensity Score Matching (PSM) method, we reveal a lower degree of market concentration under an IT framework. However, this strategy has less significant effects on inflation, unemployment, or real effective exchange rate and increases exchange rate volatility and therefore raises the systemic risk. These findings have major policy implication considering that some of our investigated countries are aspirants for the EUR adoption. Since this road requires an increased stability, especially from an exchange rate perspective, we can state that IT strategy weakens a central's bank ability to protect the national financial system against some potential currency shocks from the EUR area. Additionally, our findings indicate that central banks are more concerned with minimizing exchange rate volatility to better protect debtors with foreign currency (EUR) loans when an IT framework is not adopted. The results remain robust regardless of the approaches that we use to compute the propensity scores of IT adoption (parametric methods such as LOGIT and PROBIT or non-parametric methods such as Random Forest and Generalized Boosted).

The rest of the paper has the following structure: in Section 2, we present the literature review and its limitations. The methodology and the intuition behind the average treatment effect calculation is in Section 3. Section 4 presents the data. Section 5 gives the results and the robustness checks. Discussions are in Section 6, while Section 7 outlines the conclusions.

## 2. Literature Review

The existing literature on IT strategy has primarily investigated its effectiveness on the level of inflation and inflation volatility by comparing the outcomes before and after the adoption. In this regard, the efficiency of IT is not very clear. Some authors such as those in references [2] or [3] argued that IT was helpful to reducing the level of inflation in the short run after its implementation. Moreover, reference [4], based on 23 IT experiences and 86 non-IT countries, alongside reference [5], which is based on a sample of 36 emerging economies with 13 IT regimes, bring strong empirical arguments relating IT with a lower persistence of inflation. However, reference [6] analyzed the treatment effect of IT in industrial countries and showed that it does not exert a significant effect on inflation or its volatility in industrialized economies, while references [7] and [8] reached the same conclusion based on emerging countries' data. A mixed result on this topic has been brought to light by reference [9]. According to them, the inflation expectations after IT introduction are smaller and insignificant from a statistical point of view in comparison with non-IT countries, only in developing economies but not in industrial ones.

Regarding the impact of IT on sustainable economic growth, the opinions are also divided. Reference [10], for example, analyzed both emerging and developed economies; their findings suggest that an IT strategy really increased each country's results. However, the economic performance did not appear to be improved when compared to the case of the highly successful non-IT economies. Contradictorily, reference [7], based on a panel approach devoted to isolating the improvement in economic performance exclusively due to the IT strategy from other factors, such as endogeneity and time fixed effects, failed to find any significant relation between IT adoption and Gross Domestic Product (GDP) growth.

Another direction of research studies the relation between IT and the exchange rate. According to reference [11], IT increases exchange rate stability in developing countries but lowers it in industrial ones. On the contrary, based on a sample of developed countries, reference [12] argued that, in IT regimes, the real effective exchange rate volatility is higher. Moreover, countries that have adopted IT have a more flexible exchange rate regime than other emerging markets [13], and the monetary policy is more credible [14]. In addition, reference [15] showed that IT strategy and foreign exchange intervention policy are not sustainable, while reference [16] brought strong empirical evidence revealing that, in a flexible exchange rate environment, IT incurs a high risk of indeterminacy.

Other papers, like reference [17], suggest that IT countries are characterized by higher exposure to financial risks, in this case, central banks being less receptive to financial shocks. Different studies on the IT–financial stability nexus, such as references [18–20], point out the benefits brought by IT to reduce the sovereign debt risk especially in emerging markets. In addition, reference [21] shows the benefits of IT to pass into a more market-oriented financial system, reference [22] brings strong empirical evidence indicating that IT strategy exerts a positive effect on fiscal discipline, while reference [23] reveals that governments can gradually benefit from the adoption of IT in terms of reducing the public deficit and improve the taxation system in terms of tax revenue. Moreover, a positive but small effect of IT on business cycle is found in reference [24], the authors of which believe that business cycle synchronization depends on the monetary policy strategies adopted by the central bank.

Even if the aforementioned IT research area is comprehensive, it presents some important limitations. First, while the current literature relies upon IT's impact on inflation, inflation volatility, exchange rate volatility, the unemployment level, or economic growth, there have been no efforts to investigating how this strategy is linked with banking concentration from a treatment effect perspective. Second, most previous studies, as mentioned in the previous paragraphs, have often reported contradictory results and include, in the same sample, countries with different political regimes where monetary and fiscal policies have different features. According to reference [25], the differences in inflation, for example, between countries adopting IT (treated group) and the other countries (control group) could be due to systematic differences in some variables between IT and non-IT cases rather than due to the treatment. Third, to the best of our knowledge, there are no comparative attempts to identify, from an IT perspective, how non-EUR central banks protect the debtors exposed to currency risks. This is a crucial matter for European Union (EU) economies that are candidates for the EUR adoption, since this step requires a smooth transition without involving some possible speculative attacks on their currencies and other adverse external shocks to their banking system. In this paper, we address these issues.

## 3. Methodology

To assess the benefits brought by IT strategy, we will use the PSM algorithm proposed by the authors of reference [26]. We are particularly interested in evaluating the average effect of a treatment (IT implementation) on the treated (inflation targeting countries) about a specific outcome. We estimate the average treatment effect on the treated (ATT) as follows:

$$ATT = E\left[Y_{i,t}^1 \middle| D_{i,t} = 1\right] - E\left[Y_{i,t}^0 \middle| D_{i,t} = 1\right], \tag{1}$$

where $D_{i,t}$ is a dummy variable equal to 1 for country $i$ if the inflation targeting strategy is active in year $t$. $Y_{i,t}^1 \middle| D_{i,t} = 1$ is the value of the outcome observed for the IT country $i$ in year $t$ while $Y_{i,t}^0 \middle| D_{i,t} = 1$ is the same possible outcome achievable by a non-IT country under an IT framework. Basically, we compute the expected value of ATT as the difference between expected outcomes with and without treatment for those who were actually involved in the program. Unfortunately, $Y_{i,t}^0 \middle| D_{i,t} = 1$ is not observable so we need a proxy to make this analysis tractable. Using the average outcome of untreated observations, i.e., $E\left[Y_{i,t}^0 \middle| D_{i,t} = 0\right]$, is not recommended, considering the increased likelihood that certain factors that determine the treatment decision also determine the outcome variable of interest. Also, a major concern characterizing this approach is the self-selection bias. This phenomenon occurs when participants decide entirely for themselves whether to take part in a census or not. Following these considerations, we notice that choosing an IT strategy is not random and may depend on a series of institutional, economic, or social factors, so the problem of self-selection bias really exists.

The PSM method offers an alternative to estimate Equation (1) starting from the hypothesis that the outcomes are independent of the targeting dummy conditional on a set of covariates $X_{i,t}$. In this way, PSM gives us the possibility to select a group of non-IT countries and years to mimic a randomized

experiment in order to reducing the bias. With that in mind, we can compute the ATT from Equation (1) as

$$ATT = E\left[Y_{i,t}^1 \middle| D_{i,t} = 1, X_{i,t}\right] - E\left[Y_{i,t}^0 \middle| D_{i,t} = 0, X_{i,t}\right] \tag{2}$$

We can see from Equation (2) that the unobservable component from Equation (1), i.e., $Y_{i,t}^0 \middle| D_{i,t} = 1$ is replaced with $Y_{i,t}^0 \middle| D_{i,t} = 0$ which is known. In order to effectively compute ATT, we estimate first a standard LOGIT model to generate the propensity scores given bellow:

$$P[D_{i,t}|X_{i,t}] = \beta_0 + \beta_1 X_{i_1,t} + \beta_2 X_{i_2,t} + \ldots + + \beta_n X_{i_n,t} + \varepsilon_{i,t}, \tag{3}$$

where $X_{i,t} = \left(X_{i_1,t}, X_{i_2,t}, \ldots, X_{i_n,t}\right)$ is a matrix of covariates characterizing country $i$ in the year $t$. Generally speaking, a propensity score represents the probability that a participant in an experiment with certain features will be appointed to the treatment group (TG) or to the control group (CG). The scores allow reducing the selection bias by balancing covariates between TGs and CGs in order to match participants with multiple characteristics in a more reliable and easy way. Consequently, a matched set imply joining at least one participant in the TG and one in the CG with similar propensity scores to approximating a random trial. In our case, the propensity scores are revealing the probability that a certain country $i$ will adopt IT in year $t$. After estimating Equation (3) the propensity scores are given by:

$$p_{i,t} = \frac{e^{\hat{\beta}_0 + \hat{\beta}_1 X_{i1} + \hat{\beta}_2 X_{i2} + \ldots + \hat{\beta}_n X_{in}}}{1 + e^{\hat{\beta}_0 + \hat{\beta}_1 X_{i1} + \hat{\beta}_2 X_{i2} + \ldots + \hat{\beta}_n X_{in}}} \tag{4}$$

Furthermore, [26] showed that after determining the propensity scores, the ATT can be computed as follows:

$$ATT = E\left[Y_{i,t}^1 \middle| D_{i,t} = 1, p_{i,t}\right] - E\left[Y_{i,t}^0 \middle| D_{i,t} = 0, p_{i,t}\right] \tag{5}$$

Even though PSM usage in medicine or finance areas was extensive during the last few decades, and its advantages were clearly specified in this section, there are still disagreements regarding its effectiveness. The general consensus among the critics of the PSM relies upon the fact that the true propensity scores are unknown in empirical analysis and the estimated ones might not be accurate. Moreover, since the participants in the two groups are matched on the basis of the estimated propensity scores, the empirical analysis must take this matching into account. To overcome these limitations and to ensure robustness to our findings, in our empirical study, we are going to use different methods to estimate the propensity scores and different matching methods. For a detailed description of this methodologies, see reference [27].

## 4. The Data

In this paper we consider a balanced panel with annual data covering twelve years, from 2004 to 2015 for 11 European countries, namely Bulgaria, Croatia, Czech Republic (IT), Denmark, Hungary, Norway (IT), Poland (IT), Romania (IT), Sweden (IT), Switzerland, and the United Kingdom (IT) with non-EUR currencies. Besides the main goal of price stability, the central banks from these countries closely monitor exchange rate evolution against EUR, which, in our opinion, is an interesting fact due to the complexity of the interactions between currencies across EU countries. Moreover, it is not recommended to include countries with different political regimes where monetary and fiscal policies have different features in the same study. To overcome this problem, reference [28] included only non-IT countries with real per capita GDP at least as large as that of the poorest country with IT active in the control group. The same recommendation was made by the authors of reference [29] when studying whether banks in IT countries are less resilient and more fragile to financial risks in comparison with banks from non-IT countries.

There are currently 28 countries in the EU, and only nine economies are not in the Eurozone area. According to EU laws, countries must enter the Eurozone after reaching certain criteria regarding economic performance and financial stability, except for the UK and Denmark, which are legally exempt from adopting EUR as their national currencies. Adopting EUR as a common currency across EU eliminates exchange rate volatility, creates easy access to a monetarily unified market, and is characterized by a higher price transparency. However, the monetary policy is restricted.

While non-EUR countries can conduct monetary policies through their independent regulators, the Eurozone economies do not have this possibility. For the above-mentioned reason, we include EU member states in our database with non-EUR currencies alongside Norway and Switzerland (important partners that do not follow EU activities and regulations). By choosing this rather restrictive sample, we have ensured that our database includes only countries that are playing by the same economic or financial rules.

To perform the PSM approach, we start from a series of covariates $X_{i,t}$ that is possible to exert a significant influence on a central bank's decision to adopt the inflation targeting strategy. As a starting point, we took into account the level of central bank independence (CBI). The degree of CBI is measured based on the estimates of [30] which have been recently updated by [31]. To the best of our knowledge, this is the first attempt that took this index into consideration when investigating the effectiveness of inflation targeting strategy. Additionally, covariates such as the level of Broad Money as a percentage of GDP (BM), Political Stability Index (PSI), trade as percentage of GDP (TO), or capital account openness index (CAOI) developed by the authors of reference [32] were also included in our baseline model. These covariates have been extensively used in papers regarding inflation targeting strategy [11,21,22,33–35]. Moreover, we include in our model the level of the shadow economy (SE) provided by reference [36]. It is defined as a business activity hidden from public authorities in order to avoid paying taxes, fees, and social contributions and was found to be a key driver for the level of inflation by reference [37]. We follow reference [38] and introduce in the baseline model lagged covariates. A summary statistics is presented in Table 1.

**Table 1.** Descriptive statistics.

| Variables | Type | Min | Max | St. Dev | Avg |
|---|---|---|---|---|---|
| Lagged CBI | | 0.3070 | 0.9540 | 0.1779 | 0.7385 |
| Lagged BM | | 0.3190 | 1.8639 | 0.3684 | 0.7423 |
| Lagged PSI | | 0.0173 | 1.4183 | 0.4092 | 0.7914 |
| Lagged TO | Covariates | 0.5194 | 1.7156 | 0.3005 | 0.9835 |
| Lagged CAOI | | −0.6779 | 2.3744 | 0.8091 | 1.9773 |
| Lagged SE | | 0.0616 | 0.3058 | 0.6800 | 0.1707 |
| UNMP | | 0.0250 | 0.1770 | 0.0318 | 0.0734 |
| INFL | Outcome | −0.0114 | 0.0899 | 0.0204 | 0.0245 |
| ERV | Variables | 0.0000 | 0.1690 | 0.0409 | 0.0551 |
| REER | | 0.8158 | 1.2441 | 0.0699 | 0.9941 |
| HHI | | 0.0360 | 0.1860 | 0.0342 | 0.0968 |

Abbreviations are defined before and after Table 1.

In the first stage of the analysis, we took into consideration some additional challenger variables for the covariates, such as Control of Corruption Index, Government Effectiveness Index, Fiscal Transparency Index, Gross General Government Debt, or Foreign Direct Investment. However, we remove them after imposing a 40% threshold in absolute values for the correlation coefficients. Furthermore, to estimate the effectiveness of IT strategy, we select Unemployment (UNMP), Inflation (INFL), Exchange Rate Volatility (ERV), Real Effective Exchange Rate (REER), and Herfindahl-Hirschman Index (HHI) as outcome variables ($Y_{i,t}$). The covariates included in the PSTR model are extracted from the World Development Indicators database, while Exchange Rate Volatility is computed using daily data, and not high-frequency records, as recommended by references [39–41] when calculated realized volatility measures and portfolio strategies, mainly due to data deficiency.

## 5. Results

### 5.1. Baseline Model

The first step in our empirical analysis is to estimate the propensity scores using the LOGIT model. As we can see in Table 2, an increase in lagged central bank independence index increases the likelihood that a country will adopt an IT strategy. This result is not surprising since the central bank independence is widely considered as a precondition of the IT adoption. Furthermore, lagged Broad Money to GDP ratio is negatively associated with the probability of adopting IT strategy, as in [38]. Along the same line of argument, the stronger the lagged Trade Openness or lagged Political Stability Index, the lower the probability for a country in a certain year to adopt IT strategy. These results are similar with the previous findings reported in references [22] and [33]. Moreover, account openness has no influence on a central bank's decision to move on to an IT strategy. Finally, an increase in the level of the shadow economy in previous year will decrease probability of IT adoption in the current year. This result is very interesting since a large hidden economy might argue against IT adoption because the loss fiscal revenues undermine policy credibility. The overall performance of the LOGIT regressions is very good. According to reference [42], a pseudo-$R^2$ around 0.2 corresponds to an Ordinary Least Squares (OLS) adjusted $R^2$ of 0.7. In our case pseudo-$R^2$ equals 0.5760 and is significantly higher than the 0.2 value. This conclusion regarding the efficiency of LOGIT approach is also supported by the values of Gini and Kolmogorov–Smirnov KS coefficients, both of them showing an impressive power of discrimination for our covariates among IT and non-IT cases.

**Table 2.** Logit estimates.

| Variables | Coefficient | Standard Error | z | $P > |z|$ |
|---|---|---|---|---|
| Lagged CBI | 0.0829 | 0.0283 | 2.9300 | 0.0030 |
| Lagged BM | −0.1296 | 0.0272 | −4.7600 | 0.0000 |
| Lagged PSI | −0.0763 | 0.0205 | −3.7200 | 0.0000 |
| Lagged TO | −0.0843 | 0.0193 | −4.3600 | 0.0000 |
| Lagged CAOI | −0.6820 | 0.5346 | −1.2800 | 0.2020 |
| Lagged SE | −1.0024 | 0.2112 | −4.7500 | 0.0000 |
| Pseudo R-squared | | 0.5760 | | |
| Gini coefficient | | 0.8853 | | |
| KS test value | | 0.7727 | | |
| Observations | | 121 | | |

Once the propensity scores are estimated, we can compute, as a second step, the ATTs by applying different matching methods. First, we are going to use the nearest-neighbor algorithm, which matches each treated unit to the $n$ control units having the closest propensity scores (we consider $n = 1$, $n = 2$ and $n = 3$). In order to check the robustness of results, we will also use the radius matching approach. Nearest-neighbor algorithm presents the risk of bad matches if the nearest neighbor is very far away. This serious drawback can be bypass by imposing a tolerance threshold on the maximum score distance, which is called a caliper. Under those circumstances, reference [43] suggest an alternative for caliper matching denoted radius matching. Basically, this variant is not taking into consideration only the nearest neighbor within each radius but all of the comparison members within each radius. Following reference [22], we consider a large radius $r = 0.10$, a medium radius $r = 0.05$, and a small radius $r = 0.01$. The ATTs on inflation targeters results are presented in Table 3.

The results reported above show that there is no significant influence on the level of inflation in IT countries as compared to a non-targeter. For this reason, our findings are in line with reference [6] for developing economies and contradict the ones reported in references [7] and [8]. Similar results can also be drawn for unemployment or real effective exchange rate. Even though the signs of the coefficients are pointing out for IT countries, a higher inflation, higher unemployment, or higher real effective exchange rate for some certain matching methods, it lacks proper evidence to support these

findings, as we will see in the next section. However, we have achieved robust evidence indicating that exchange rate volatility is higher in IT countries, the difference among them being around 6 p.p. on a yearly basis. Finally, our results indicate, based on HH Index difference, that the banking system in IT countries is less concentrated in comparison with banks from non-IT countries.

**Table 3.** Average treatment effect on the treated (ATT) estimates based on Logistic Propensity Scores.

| Matching Method | Nearest Neighbor Matching | | | Radius Matching | | |
|---|---|---|---|---|---|---|
| | *n*=1 | *n*=2 | *n*=3 | *r*=0.01 | *r*=0.05 | *r*=0.1 |
| UNPL | 0.0161 | 0.0148 | 0.0159 * | 0.0206 *** | 0.0039 | 0.0067 |
| INFL | 0.0032 | 0.0047 | 0.0048 | 0.0016 *** | 0.0101 *** | 0.0065 |
| ERV | 0.0651 *** | 0.0638 *** | 0.0634 *** | 0.0793 *** | 0.0677 *** | 0.0676 *** |
| REER | 1.8190 | 2.0801 | 2.3189 | 0.8819 *** | 2.1723 *** | 0.1397 |
| HDI | −0.0205 ** | −0.0198 * | −0.0211 * | −0.0244 * | −0.0221 ** | −0.0192 * |

***, ** and * indicates significant at 1%, 5% and 10% level.

### 5.2. Robustness of Results

In this section, we will put our results to a series of robustness checks others than the matching methods. First of all, we use different specifications in order to estimate the propensity scores. Afterward, we will include some additional variables in the baseline model, and finally, we will replace lagged covariates with contemporary ones when explaining the probability of IT adoption via a LOGIT representation.

#### 5.2.1. Different Specifications for Propensity Scores

In regard to model selection, although the treatment effects are estimated based on different matching methods, the propensity scores are estimated using one model with a fixed set of covariates. In these subsection different methods for propensity scores estimations such as PROBIT, Random Forest [44], and Generalized Boosted [45,46] are performed in order to compute the ATT.

The results reported in Table 4 are confirming the aforementioned conclusions stated in Section 5.1, according to which IT has no robust impact on inflation, unemployment and the real effective exchange rate, but it increases exchange rate volatility and decreases market concentration.

**Table 4.** Robustness of results.

| Matching Method | Nearest Neighbor Matching | | | Radius Matching | | |
|---|---|---|---|---|---|---|
| | *n*=1 | *n*=2 | *n*=3 | 0.01 | 0.05 | 0.1 |
| Probit Model | | | | | | |
| UNPL | 0.0173 | 0.0174 * | 0.0170 * | 0.0154 *** | 0.0169 | 0.0074 ** |
| INFL | 0.0044 | 0.0054 | 0.0055 | 0.0174 *** | 0.0158 *** | 0.0104 *** |
| ERV | 0.0653 *** | 0.0632 *** | 0.0616 *** | 0.0604 *** | 0.0652 *** | 0.0661 *** |
| REER | 1.8139 | 2.4451 | 2.5578 | 2.667 *** | 1.7971 *** | 0.5073 |
| HDI | −0.0223 ** | −0.0218 * | −0.0265 | −0.0227 * | −0.0256 ** | −0.0193 * |
| Generalized Boosted Modeling | | | | | | |
| UNPL | 0.0274 * | −0.0224 | −0.0462 | 0.0332 ** | −0.0224 | −0.0462 |
| INFL | 0.0063 | 0.0102 | 0.0074 | 0.0063 * | 0.0102 * | 0.0074 |
| ERV | 0.0674 *** | 0.0611 *** | 0.0584 *** | 0.0689 *** | 0.0703 *** | 0.0576 *** |
| REER | 2.8419 | 1.7302 | 2.0576 | 2.7612 | 2.7562 | 4.1239 |
| HDI | −0.0251 ** | −0.0281 * | −0.0233 | −0.0250 * | −0.0203 ** | −0.0188 * |

**Table 4.** *Cont.*

| Matching Method | Nearest Neighbor Matching | | | Radius Matching | | |
|---|---|---|---|---|---|---|
| | *n*=1 | *n*=2 | *n*=3 | 0.01 | 0.05 | 0.1 |
| | Random Forest | | | | | |
| UNPL | 0.0250 | 0.0257 ** | 0.0096 | 0.0238 * | −0.0410 | −0.0331 |
| INFL | 0.0064 | 0.0056 | −0.0045 | 0.0063 | 0.0102* | 0.0074 |
| ERV | 0.0672 *** | 0.0676 *** | 0.0648 *** | 0.0648 *** | 0.0655 *** | 0.0509 *** |
| REER | 2.8142 | 3.0167 | 7.8882 * | 2.8345 | 3.0489 * | 3.541 |
| HDI | −0.0199 ** | −0.0232 * | −0.0203 | −0.0255 ** | −0.0218 * | −0.0176 * |

***, ** and * indicates significant at 1%, 5% and 10% level.

### 5.2.2. Additional Covariates in Baseline Model

One reason for higher exchange rate volatility in IT targeters is that these countries usually do not follow the goals of exchange rate stability and price stability simultaneously, and the volatility of exchange rate is high because the rates are not pegged. For this reason, it is possible that additional covariates such as lagged exchange rate regime (ERR) or lagged inflation rate (IR) in the first level to tell a different story for the average treatment effect on the treated. ERR is classified according to the level of freedom by reference [47] in four categories—pegged regimes, crawling pegs, managed floats, and freely floating regimes. The LOGIT results are in Table 5.

**Table 5.** LOGIT estimates for baseline model with lagged ERR and lagged IR as covariates.

| Variables | Coefficient | Standard Error | *z* | *P > |z|* |
|---|---|---|---|---|
| Lagged CBI | 0.0581 | 0.0301 | 1.9320 | 0.0533 |
| Lagged BM | −0.1169 | 0.0267 | −4.3820 | 0.0000 |
| Lagged PSI | −0.0721 | 0.0205 | −3.5140 | 0.0004 |
| Lagged TO | −0.0681 | 0.0200 | −3.4030 | 0.0007 |
| Lagged CAOI | −1.0650 | 0.6406 | −1.6630 | 0.0964 |
| Lagged SE | −0.9760 | 0.2179 | −4.4790 | 0.0000 |
| Lagged IR | 0.3695 | 0.2539 | 1.4550 | 0.1456 |
| Lagged ERR | −1.7213 | 171.90 | −0.0100 | 0.9920 |
| Pseudo R-squared | | 0.5835 | | |
| Gini coefficient | | 0.8964 | | |
| KS test value | | 0.8151 | | |
| Observations | | 121 | | |

Abbreviations ERR and IR are defined before Table 5.

Once again, as we can see in Table 6, the estimates presented in Tables 2 and 3 remains robust. Nevertheless, we fail to identify a significant coefficient relating lagged inflation rate to the likelihood of IT adoption. This result invalidates the conclusions stated in references [21,22,33], who find negative relations and argue that IT should be adopted only after some preconditions regarding the level of inflation are satisfied. However, in previous studies the same authors find a negative relationship between the probability of IT adoption and GDP growth, which contradicts the relationship between IT and inflation level.

**Table 6.** ATT estimates based on extended LOGIT representation.

| Matching Method | Nearest Neighbor Matching | | | Radius Matching | | |
|---|---|---|---|---|---|---|
| | *n*=1 | *n*=2 | *n*=3 | *r*=0.01 | *r*=0.05 | *r*=0.1 |
| UNPL | 0.0146 | 0.0158 | 0.1153 | −0.0093 *** | 0.0082 | 0.0090 |
| INFL | 0.0078 | 0.0070 | 0.0062 | 0.0048 *** | 0.0108 | 0.0078 |
| ERV | 0.0626 *** | 0.0605 *** | 0.0601 *** | 0.0607 *** | 0.0584 *** | 0.0608 *** |
| REER | 1.9555 | 2.2924 | 2.2127 | −4.4063 *** | 1.2858 | 0.9316 |
| HDI | −0.0187 * | −0.0165 * | −0.0277 ** | −0.0272 * | −0.0206 | −0.0177 * |

***, ** and * indicates significant at 1%, 5% and 10% level.

### 5.2.3. Robustness Checks with Contemporary Variables

The final step of our robustness analysis is to estimate the Equation (3) by using non-lagged variables. The results are presented in Table 7:

**Table 7.** LOGIT estimates.

| Variables | Coefficient | Standard Error | *z* | *P* > |*z*| |
|---|---|---|---|---|
| CBI | 9.2956 | 3.2423 | 2.8670 | 0.0041 |
| BM | −0.1427 | 0.0318 | −4.4890 | 0.0000 |
| PSI | −7.7740 | 2.2222 | −3.4980 | 0.0005 |
| TO | −0.0921 | 0.0216 | −4.2700 | 0.0000 |
| CAOI | −0.6811 | 0.6228 | −1.0940 | 0.2741 |
| SE | −1.1085 | 0.2440 | −4.5430 | 0.0000 |
| Pseudo R-squared | | 0.5940 | | |
| Gini coefficient | | 0.9013 | | |
| KS test value | | 0.7818 | | |
| Observations | | 121 | | |

As we can see in Table 8, the robustness checks validate the fact that the impact coefficients highlighted in in Tables 2 and 3 retain their signs and statistical significance. The stronger the central bank independence, the stronger the probability for a country in a certain year to adopt IT strategy. Furthermore, variables such as Broad Money to GDP, Trade Openness, Political Stability Index, or the Shadow Economy are negatively associated with the probability of adopting IT strategy.

**Table 8.** ATT estimates based on lagged ERR and lagged IR.

| Matching Method | Nearest Neighbor Matching | | | Radius Matching | | |
|---|---|---|---|---|---|---|
| | *n*=1 | *n*=2 | *n*=3 | *r*=0.01 | *r*=0.05 | *r*=0.1 |
| UNPL | 0.227 ** | 0.0206 ** | 0.0207 *** | 0.0357 *** | 0.0138 *** | 0.226 ** |
| INFL | 0.0053 | 0.0054 | 0.0051 | 0.0175 *** | 0.0048 | 0.0046 |
| ERV | 0.0629 *** | 0.0614 *** | 0.0621 *** | 0.0622 *** | 0.0587 *** | 0.0630 *** |
| REER | 3.1097 | 3.3479 | 3.1358 | 0.8448 *** | 0.8104 | 3.0917 |
| HDI | −0.0218 ** | −0.0203 * | −0.0201 * | −0.0255 * | −0.0241 * | −0.0189 |

***, ** and * indicates significant at 1%, 5% and 10% level.

## 6. Discussion

Our results highlight that IT brings no benefit to society in terms of inflation, unemployment, or real effective exchange rate. However, IT is associated with an increase in exchange rate volatility and a decrease in banking concentration. In our view, these findings have major implications. Even though price stability is the main goal stated in a central bank's agenda, in an IT environment, the association with other secondary goals such as employment and economic growth maximization is detrimental for

financial stability. As we can see in the reported results, this association is not followed by a decrease in inflation level or by an increase in employment or economic growth. This fact may lead to a decrease in a central's bank credibility among economic agents, making it harder to control inflation or other financial shocks. However, in an IT regime, banking concentration is lower in comparison to non-IT countries. This translates in lower credit costs for economic agents and higher diversity among banking products, which is beneficial for the economy.

An interesting conclusion drawn from our analysis is that, in non-IT countries, central banks are trying to protect their population, especially those with foreign currency (EURO) loans, by minimizing exchange rate volatility in comparison with IT countries. This result is very interesting and shows us the inferiority of IT when it comes to achieve different social or financial goals, other than price stability. A possible explanation relies on the fact that exchange rate volatility is linked to the monetary policy rules in non-IT countries. This hypothesis was launched by the author of reference [48,49], and according to him, systemic risk is substantial higher in the non-EURO area where loans denominated in EURO contribute significantly to this type of risk. Foreign currency loans that were sold during the economic boom have become the credit product with the highest rate of default during the recent financial turmoil [49]. The interest spread was an important factor that made clients accept foreign exchange risk and buy EURO loans at significantly lower costs. Under those circumstances, there are reasons for us to think that the central banks in non-IT countries were concerned to keep this volatility at a minimum level in order to protect the debtors from exchange rate risk and, as a consequence to maintain financial stability.

To test this hypothesis, we will study the link between exchange rate volatility and the percentage of EURO loans in total loans using Granger causality test in a panel data framework. The data regarding EURO loans were collected from each central bank's website, except for Norway, where the data set was not available.

As we can see in Table 9 the proportion of EUR loans in total loans was a driving factor for each central's bank behavior in respect to exchange rate volatility in non-IT countries. This result is indicating that in IT strategy the exchange rate stability is not considered as a priority, thus complicating the integration into the Eurozone for IT countries with non-EUR currencies.

**Table 9.** Granger causality panel data.

| Null Hypothesis | F-statistic | *P*-value |
|---|---|---|
| IT countries | | |
| ERV does not Granger cause EURO loans % Total Loans | 0.61193 | 0.4387 |
| EURO loans % Total Loans does not Granger cause ERV | 0.31793 | 0.5760 |
| Non-IT countries | | |
| ERV does not Granger cause EURO loans % Total Loans | 0.05387 | 0.8175 |
| EURO loans % Total Loans does not Granger cause ERV | 5.33127 | 0.0254 |

These conclusions might be interesting for international institutions such as the European Commission, World Bank, or International Monetary Fund (IMF). Those institutions may guide particular IT countries to implement policies devoted to reducing the level of inflation given alongside other goals when IT strategy is active. However, the World Bank or IMF should be very careful about each central bank's strategy in a particular country especially when that country calls for a bailout program. In the years following the bailout program, governments will have to significantly increase the public debt to pay off debts [50], which will cause an increase in inflation and an escalation of macroeconomic imbalances.

## 7. Conclusions

This paper has carried out an extensive assessment regarding the effectiveness of the inflation targeting strategy with the help of the Propensity Score Matching approach. By using non-EUR

European countries that adopt an IT and non-EUR European countries that do not adopt an IT for the period of 2005–2015, we can highlight several contributions to literature. First of all, by using a LOGIT model, we identify broad money as a percentage of GDP, political stability index, trade as a percentage of GDP, alongside central bank independence and the level of the shadow economy (as a novelty) as the main drivers for IT adoption. Second, the estimates suggest that the IT adoption brings no benefit to society in terms of inflation, unemployment or real effective exchange rate. Third, inflation targeting strategy increases exchange rate volatility and thus raises the systemic risk but can be associated with a less concentrated banking systems. In addition, based on a Granger causality test, we reveal that in non-IT countries in comparison with IT countries, central banks are more concerned with minimizing the exchange rate volatility in order to better protect the debtors with foreign currency (EURO) loans. Fourth, as a plus for the IT regime, the banking concentration is lower, translating to lower credit costs for economic agents and higher diversity among banking products, which is beneficial for the economy. The results remain robust regardless of the approaches that we use in propensity scores calculation or the matching method. However, this research presents some limitations, such as a mixed logit model for panel data utilization for propensity score estimation, or different techniques for ATT calculation, such as regression discontinuity design or difference in difference regression. All this, alongside a larger sample for estimating the impact of IT in a mixed model with government fiscal policies, will be left for further research. Overall, these findings show us the inferiority of IT when it comes to achieving different social or financial goals, other than price stability.

**Author Contributions:** Conceptualization: A.I.S. and B.A.D.; Data curation: B.A.D.; Formal analysis: A.I.S.; Investigation: A.I.S. and B.A.D.; Methodology: A.I.S.; Validation: A.I.S.; Project administration: B.A.D.; Writing—original draft: A.I.S. and B.A.D.; Writing—review and editing: A.I.S. and B.A.D.

**Funding:** This research received no external funding.

**Conflicts of Interest:** The authors declare no conflicts of interests.

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
