# Peer review of "Assessing the Sustainability of Inflation Targeting: Evidence from EU Countries with Non-EURO Currencies"

_sustainability, doi:10.3390/su11205654_

Round 1

Reviewer 1 Report

The article is very interesting and relevance for the knowledge base. The significance value of the paper is the empirical econometric analysis which concentrates on the impact of inflation targeting on specific macroeconomic variables. It adheres to the Journal’s standards.

The recommended changes are described as follows:

1) When it comes to the structure of the paper, it is necessary to improve the specific sections. First, the introduction should in general introduce to the topic, while now it is combined with the previous research analysis, which should be presented in the next - literature review section. Second, the Author should add the literature review section and present all the necessary theoretic knowledge about the undertaken topic. It can add the previously obtained research results.

2) The Methodology section should also be extended to present readers how this econometric methods work and argue why the Author chose these methods. What are the main strengths and weaknesses to use these methods.

3) The conducted research are interesting and significance but presentation of their results are insufficient. The Author need to evaluate the obtained results not only be presenting quantitative data, but it should try to analysis what are the main conclusions from them. Now there is only presentation of the data without any significance evaluation. Especially not only from econometric point of view but from financial aspect – central banking and monetary policy belongs to the discipline of economics and finance.

4) In row 231 there is a grammatical mistake – There is a beginning of the sentence “However,” but there are no any further text.

5) As it was mentioned before. The section Discussion and Conclusions should be extended and corrected for sure. There no any logical conclusions and assessment of the conducted research. Usually, the conclusions section also include some recommendation for future research or some research questions that might be a basis for the further in-depth research. Very low level of inference. Without these changes the quality of the paper is insufficient to be published in the Journal.

However, the article is interesting and undertake important topic for the central banks’ strategy and its impact on macroeconomic variables. That’s why I recommend to make major changes in the paper and send it to the re-review later on.

Author Response

1) When it comes to the structure of the paper, it is necessary to improve the specific sections. First, the introduction should in general introduce to the topic, while now it is combined with the previous research analysis, which should be presented in the next - literature review section. Second, the Author should add the literature review section and present all the necessary theoretic knowledge about the undertaken topic. It can add the previously obtained research results.

Response from authors: Thank you very much for this comment. We revised our initial version of the paper according to your recommendations. More specifically, in the Introduction section we describe more extensively the main features of IT strategy and pointing out the main contribution of our paper. In the Literature review section we add some additional studies investigating the relation between IT and public debt, business cycle synchronization, and tax revenues respectively. Moreover, we bring to light the gaps from literature and how we overcome these issues.

2) The Methodology section should also be extended to present readers how these econometric methods work and argue why the Author chose these methods. What are the main strengths and weaknesses to use these methods?

Response from authors: Thank you very much for this comment. We revised our initial version of the paper according to your recommendations. More specifically, we described more accurately the PSM methodology while presenting at the same time its limitations alongside the solutions that we offer in order to assess the robustness of results.

3) The conducted research is interesting and significance but presentation of their results are insufficient. The Author need to evaluate the obtained results not only be presenting quantitative data, but it should try to analysis what are the main conclusions from them. Now there is only presentation of the data without any significance evaluation. Especially not only from econometric point of view but from financial aspect – central banking and monetary policy belongs to the discipline of economics and finance.

Response from authors: Thank you very much for this comment. However, in this case, the Results section is devoted to presenting the results along with robustness checks from a statistical point of view, while economic significance evaluation is left for the Discussion section.     

4) In row 231 there is a grammatical mistake – There is a beginning of the sentence “However,” but there are no any further text.

Response from authors: Thank you very much for this comment. We fixed this issue.

5) As it was mentioned before. The section Discussion and Conclusions should be extended and corrected for sure. There no any logical conclusions and assessment of the conducted research. Usually, the conclusions section also includes some recommendation for future research or some research questions that might be a basis for the further in-depth research. Very low level of inference. Without these changes the quality of the paper is insufficient to be published in the Journal.

Response from authors: Thank you very much for this comment. We extended the Discussion and Conclusions by presenting the economic significance of results, policy implications and offering in the same time some recommendation for future research or some research questions.

Reviewer 2 Report

Title: Assessing the sustainability of inflation targeting. Evidence from EU countries with non-EURO currencies.

At first, I think that a different section with the literature review will enhance the value of the study. The part of literature review should be clear and more extended in order to be understandable the differences of the present study. The authors should explain better the specific features of non-EURO EU countries with respect to EU countries This explanation will fully justify the choice of these countries. Why the authors choose this time period for their research? Why the authors focus on simple Logit and not on the Mixed Logit (see Train 2003, Tsagkanos 2007)? The analysis will be solid incorporating and testing theoretical hypotheses.

I think that a revised version with the abovementioned concerns could be a contribution to the literature.

Literature

Train, K., 2003. Discrete Choice Methods with Simulation. Cambridge University Press, New York.

Tsagkanos G. A. (2007), “A bootstrap – based minimum bias maximum simulated likelihood estimator of Mixed Logit.”, Economics Letters, 96, 282 – 286.

Author Response

Response from authors: Thank you very much for this comments. We revised our initial version of the paper according to some of your recommendations. More specifically:

In the Introduction section we described more extensively the main features of IT strategy and pointing out the main contribution of our paper. In the Literature review section we add some additional studies investigating the relation between IT and public debt, business cycle synchronization, and tax revenues respectively. We provided an in-depth discussion regarding the specific features of non-EURO EU countries with respect to EU countries. 2005–2015 was the largest time period with all variables available. In ITs literature, the LOGIT or PROBIT models are chosen in order to calculate the propensity scores. So we follow this path and use a simple LOGIT approach for our baseline model. However, some additional estimation methods for propensity scores such as PROBIT or Random Forest were employed in the robustness analysis. With all that, your recommendation is interesting and will definitively be considered in further research papers. Thank you very much for your recommendation. However, we have pointed out in data description the reason for choosing the variables and explained in the Results section the expected signs in comparison with present studies.

Round 2

Reviewer 1 Report

The Authors made sufficient changes in the paper.

I recommend to publish the article in the current form.